# Epithelial-to-Mesenchymal Transition-Derived Heterogeneity in Head and Neck Squamous Cell Carcinomas

**DOI:** 10.3390/cancers13215355

**Published:** 2021-10-26

**Authors:** Philipp Baumeister, Jiefu Zhou, Martin Canis, Olivier Gires

**Affiliations:** 1Department of Otorhinolaryngology, University Hospital, LMU Munich, D-81377 Munich, Germany; Jiefu.Zhou@med.uni-muenchen.de (J.Z.); martin.canis@med.uni-muenchen.de (M.C.); 2Clinical Cooperation Group “Personalized Radiotherapy in Head and Neck Cancer”, Helmholtz Zentrum München, German Research Center for Environmental Health GmbH, D-85764 Neuherberg, Germany

**Keywords:** head and neck squamous cell carcinoma, (partial) epithelial-to-mesenchymal transition, therapy resistance

## Abstract

**Simple Summary:**

Head and neck squamous cell carcinomas (HNSCC) are common malignancies with considerable morbidity and a high death toll worldwide. Resistance towards multi-modal therapy modalities composed of surgery, irradiation, chemo- and immunotherapy represents a major obstacle in the efficient treatment of HNSCC patients. Patients frequently show nodal metastases at the time of diagnosis and endure early relapses, oftentimes in the form of local recurrences. Differentiation programs such as the epithelial-to-mesenchymal transition (EMT) allow individual tumor cells to adopt cellular functions that are central to the development of metastases and treatment resistance. In the present review article, the molecular basis and regulation of EMT and its impact on the progression of HNSCC will be addressed.

**Abstract:**

Head and neck squamous cell carcinomas (HNSCC) are common tumors with a poor overall prognosis. Poor survival is resulting from limited response to multi-modal therapy, high incidence of metastasis, and local recurrence. Treatment includes surgery, radio(chemo)therapy, and targeted therapy specific for EGFR and immune checkpoint inhibition. The understanding of the molecular basis for the poor outcome of HNSCC was improved using multi-OMICs approaches, which revealed a strong degree of inter- and intratumor heterogeneity (ITH) at the level of DNA mutations, transcriptome, and (phospho)proteome. Single-cell RNA-sequencing (scRNA-seq) identified RNA-expression signatures related to cell cycle, cell stress, hypoxia, epithelial differentiation, and a partial epithelial-to-mesenchymal transition (pEMT). The latter signature was correlated to nodal involvement and adverse clinical features. Mechanistically, shifts towards a mesenchymal phenotype equips tumor cells with migratory and invasive capacities and with an enhanced resistance to standard therapy. Hence, gradual variations of EMT as observed in HNSCC represent a potent driver of tumor progression that could open new paths to improve the stratification of patients and to innovate approaches to break therapy resistance. These aspects of molecular heterogeneity will be discussed in the present review.

## 1. Introduction

Head and neck squamous cell carcinoma (HNSCC) have an incidence of approximately 600,000 new cases worldwide per year and represent the sixth most frequent cancer type [1,2,3]. Overall survival rates remain rather poor despite multi-modal treatment approaches composed of surgical removal of the primary tumor and cervical lymph nodes, (neo)adjuvant radio(chemo) therapy (RCT), and targeted therapy, including epidermal growth factor receptor (EGFR)- and programmed cell death protein 1 (PD1) and programmed death ligand (PD-L1)-specific biologicals. Major parameters related to a poor outcome of human papillomavirus (HPV)-/p16-negative HNSCC are the presence of lymph node metastases and local or loco-regional recurrences, which are a reflection of potent therapy resistance in these malignancies [4]. High mutational burden in gene coding regions is correlated with poor prognosis of HNSCC [5,6] and the degree of genetic heterogeneity is a measure for the clinical outcome of patients. High values of mutant allele tumor heterogeneity (MATH) correlate with reduced survival [7]. Generally, intra- and inter-tumor heterogeneity (ITH) entails major obstacles in the treatment of malignant tumors. A solid cancer contains several billion tumor and stromal cells at the time of diagnosis, including many genetically and epigenetically distinct cell clones exhibiting diverse phenotypes and functional features. Strong ITH has a significant prognostic value across many cancer types where it associates with poorer clinical outcome and is particularly pronounced in HNSCC [8,9].

From a tumor evolutionary standpoint, the mucosal lining of the upper aerodigestive tract is subject to field cancerization. Predominantly in HPV-negative HNSCC, epithelial cells are chronically exposed to tobacco-contained carcinogens, frequently in combination with alcohol consumption and typically for decades. Alternatively, gastroesophageal reflux, betel nut, and airborne agents can also induce molecular changes leading to pre-malignancies such as Barret’s esophagus, amongst others [4,10]. Stochastic in principle, some carcinogen-induced (epi)genetic alterations may confer survival and/or proliferative advantages to the affected cell and its offspring. Therefore, these cells will sooner or later replace neighboring cells lacking these advantages, while being further exposed to carcinogenic impact. Thus, different areas containing a variety of premalignant variants of mucosal cells develop, termed (preneoplastic) *fields*. Some variants eventually complete malignant transformation to become the originator of a primary squamous cell carcinoma [11,12,13]. Because of the multilocular nature of field cancerization, HNSCC patients are often diagnosed with syn- or metachronous second primary carcinomas. Due to their invisibility to the naked eye and conventional histopathological examination techniques, it can be assumed that premalignant fields of origin are seldomly surgically excised and/or irradiated in toto. Therefore, local recurrences in terms of second field tumors arising from preneoplastic mucosa left in situ remain frequent even when histopathologically clear resection margins were achieved during the surgical resection of the primary carcinoma [14,15,16].

The selective pressure promoting the outgrowth of the fittest premalignant clones may be less pronounced during early stages of carcinogenesis. However, a growing scarcity of resources such as nutrients and oxygen, physical barriers such as the basement membrane or progressive perilesional fibrosis, and/or increasing immune surveillance represent powerful restrictors of malignant growth. Genomic heterogeneity and phenotypical diversity already present and further increasing during carcinogenesis are excellent prerequisites to overcome these limitations. Although not purposefully produced, those clones able to overcome specific boundaries arising at a specific time point and at a specific location of tumor growth will prevail and be selected [17]. Gerlinger and colleagues impressively demonstrated the branched evolution of clear-cell carcinoma cells obtained from different localizations within the primary tumor, as well as from multilocular metastases. Tissue samples showed mutations common to all localizations, mutations exclusively shared by biopsies within the primary tumor, as well as mutations shared by distant metastases only [18].

Despite the appealing logic of Darwinian principles applied to oncology, it remains largely unknown how much of the genomic heterogeneity found in different cancers has functional consequences [19]. It may well be that driver gene mutations are responsible only for fundamental aspects of malignant growth such as replicative immortality, evasion of growth suppressors, and resistance to cell death. Other hallmarks of cancer such as invasive and metastatic growth, deregulated metabolism, inflammatory responses, immune escape, and therapy resistance [20] may primarily be caused by altered signaling and epigenetically regulated programs such as EMT, hypoxic response, or xenobiotic defense. Notably, the expression of these programs and their vast effects on cellular phenotypes cannot be detected by sequencing cancer genomes but relies on transcriptomic and proteomic approaches. Eventually, a combination of multi-omic approaches that comprehensively tackle DNA mutations, epigenetic regulation, and protein expression and activity are central to a deepened understanding of HNSCC progression. For instance, such a proteogenomic approach recently uncovered a far more important role of the expression of EGFR ligands than of the receptor itself in determining EGFR-targeted therapy responses, thus providing valuable new molecular insights [21].

In this review, we focus on the role of EMT in the development of heterogeneity and therapy resistance.

## 2. Sources of Phenotypic Diversity in HNSCC

Cancer cells differ in numerous phenotypic and functional aspects, including their morphology, proliferation, metabolism, motility, invasiveness, their ability to induce angiogenesis and to evade immune responses. Phenotypic ITH is frequently accompanied by excessive genetic heterogeneity. Impaired DNA repair and errors during replication and/or chromosome segregation are common causes for a wide range of lesions, such as point mutations, insertions and/or deletions, as well as chromosomal rearrangements and alterations of the copy number of entire chromosomes [22]. Thus, cancer cells accumulate somatic mutations and show losses and/or gains of chromosomes and segments. Although the tolerance towards genomic instability appears limited, even cancer cells that show copy number changes in large proportions of their genome can survive and proliferate extensively. Next-generation DNA sequencing approaches have demonstrated that HNSCC accumulate on average 130 mutations in exome regions and that 75% of these mutations result in non-silent changes in coding regions [6]. In HPV-negative HNSCC, most frequent mutations and chromosomal alterations occurred in the driver genes *P53* (84%), *CDKN2A* (58%), *PIK3CA* (34%), *CCND1* (31%), *EGFR* (15%), *MYC* (14%), *NOTCH1* (14%), and *FGFR1* (10%) [5]. Knowledge of these frequent mutations is essential and contributes to the definition of potential targets for therapy. For example, gene amplifications of EGFR can promote EGFR-dependent signaling and qualify this receptor tyrosine kinase as a therapeutic target in HNSCC treatment, although predominantly for palliative treatment [4,23].

An additional source of ITH is epigenetic alterations regulating the transcriptional activity of genes via chromatin remodeling, DNA methylation, and non-coding RNAs. In HNSCC, changes in nuclear size, prominent nucleoli, dense hyperchromatic DNA, and a high nuclear-cytoplasmic ratio indicate profound alterations in chromatin structure and function [24]. It has been proposed that epigenetic modifications, particularly heritable methylation patterns, are more likely to initiate neoplastic growth than somatic mutations [25,26]. In fact, methylation patterns across carcinomas are so specific that they allow differentiating HNSCC-derived lung metastasis from primary lung carcinoma despite the huge similarity at the cellular level [27]. Altered regulation of transcriptional activity by promoter-methylation was shown for genes with central roles in cell cycle arrest, DNA-repair, apoptosis, cytoskeleton organization, cellular adhesion, and migration.

Prominent genes can be functionally altered by multiple mechanisms. As a good example, the gene for cyclin-dependent kinase inhibitor 2A (*CDKN2A*), which codes for the two cell cycle inhibitors p14 and p16, is not only prone to homozygous deletions in HNSCC. Additionally, *CDKN2A* becomes silenced via histone 3 lysine 9 hypermethylation and lysine 4 hypomethylation, resulting in reduced gene transcription that can be counteracted using demethylating agents [28,29,30].

Non-coding microRNAs (miRNAs) compose yet another source of regulation involved in the post-transcriptional modulation of genes, which can be used as biomarkers in HNSCC [31]. A five-miRNA signature was extracted from *n* = 85 patients and was validated in a second cohort of *n* = 77 patients as a strong and independent prognosticator for recurrence and survival in HPV-negative HNSCC. These miRNAs were partly known from other solid tumors, including breast, pancreatic, and non-small cell lung cancer, and were previously associated with hypoxic response, mitochondrial metabolism, survival, and immune responses. Three miRNAs of the five-miRNA signature (namely hsa-miR-6508-5p, hsa-miR-4306, and hsa-miR-7161-3p) had not been related to cancer previously. These miRNAs have been linked to prominent tasks such as p53 signaling, DNA double-strand break repair, pre-*NOTCH* expression, and senescence-associated pathways [32].

In summary, cancer cells exploit diverse physiological cellular programs, thereby changing their phenotype and functionality, including the induction of gradual forms of EMT. Currently available data do not suggest a major contribution of specific mutations in initiating or orchestrating this reversible transition to more mesenchymal features. However, somatic mutations and differential epigenetic modulation can alter signaling pathways and, thus, influence the equilibrium either towards or against the activation of EMT. The progressive transition from epithelial to mesenchymal phenotypes can be categorized into three variants termed EMT type 1 to 3. EMT type 1 has a crucial role during embryogenesis at the stage of gastrulation and germ layer definition, whereas EMT type 2 is instrumental in wound healing. In cancer, EMT type 3 is associated with malignant progression conferring multiple traits of high-grade malignancy to carcinomas, including tumor-initiating potential, therapy resistance, as well as locally invasive and metastatic growth [33,34,35,36,37]. It must, however, be stated that metastasis formation in the absence of overt signs of EMT have been reported in other entities [38,39]. Therefore, the role of EMT as an indispensable contributor to cancer progression remains a matter of debate [36,40,41]. Diverse degrees of EMT in HNSCC, their regulation and implications will be discussed in the following chapter.

## 3. EMT in HNSCC

Over the past 15 years, PubMed entries with the search term “HNSCC EMT” have gradually increased every year and have accumulated to over 500 publications in 2021. Partial forms of EMT (i.e., pEMT) have been implicated in several central aspects of cancer progression and emerged as a prognostic marker with functional relevance in HNSCC. Malignant cells from various entities, including HNSCC, typically do not undergo a full and irreversible EMT. Rather, malignant cells adopt mesenchymal markers and functions while only reducing or retaining epithelial characteristics. Despite this notion, we will use the term EMT in the present review when globally addressing EMT but use the term pEMT when specifically acknowledging its incomplete nature and marker signatures in HNSCC.

Functionally, EMT enhances the invasive capacity of tumor cells and thereby supports local invasion in tumor-surrounding tissue to initiate minimal residual disease (MRD), which represents a source of recurrence [42]. Additionally, EMT can enhance tumor cell invasion into lymphatic and blood vessels, thus fostering nodal and distant metastases (Figure 1) that strongly impact on the clinical outcome of patients [43,44].

Both MRD and systemic cancer cells represent essential hurdles for an efficient therapeutic treatment of HNSCC. These clinically challenging occult tumor cells are highly problematic owing to a potentially stem-like and dormant phenotype associated with enhanced therapy resistance. Cancer stemness features are fostered via EMT and metabolic changes that are mostly associated with a rewiring towards glycolysis at the expense of oxidative phosphorylation [33,36,37,45,46,47,48]. In the following, diverse aspects of EMT in HNSCC will be summarized. These aspects include molecular mechanisms of EMT regulation, the functional implications of EMT in disease progression, the definition of EMT gene signatures for prognostic and predictive purposes, and the role of EMT in treatment resistance and stemness.

### 3.1. Regulation of EMT in HNSCC

Several signaling pathways and six core EMT-inducing transcription factors orchestrate the regulation of EMT in malignant cells [34,46,49,50]. Major receptor-mediated pathways and the executing down-stream effectors involved in the regulation of EMT in HNSCC will be described in the following paragraph.

#### 3.1.1. TGF-β1-Dependent EMT Regulation

The transforming growth factor-β1 (TGF-β1)-induced pathway is amongst the best-explored signaling pathway involved in EMT regulation in HNSCC [51]. TGF-β1 induces an activating auto-phosphorylation of TGF-β receptors 1 and 2 (TGFβR1/2) upon dimerization. Phosphorylated TGFβR1/2 present docking sites to SMAD transcription factors (human homologs of the Mad and Sma proteins identified in *Drosophila* and *C. elegans*). Thereby, different combinations of SMAD homo- and heterodimers become activated via phosphorylation, form trimers with a common partner SMAD (co-SMAD), translocate into the nucleus, and regulate the activation or repression of target genes through the recruitment of transcriptional activators or repressors, respectively [52,53,54] (Figure 2A).

TGF-β1 activates EMT in HNSCC through a SMAD-dependent pathway [51,55,56,57,58] and is instrumental in the formation of tumor buds that detach from primary tumors in oral SCC (OSCC) via activation of ZEB1 and paired-related homeobox 1 protein PRRX1 [59]. Importantly, TGF-β1-associated tumor budding correlated with nodal involvement and predicted poorer overall survival (OS) [59]. Involvement of TGF-β1 in the induction of nodal metastases in HNSCC was corroborated by numerous publications along with regulatory mechanisms of EMT activation or repression [57,60,61,62,63,64]. TGF-β1 acts in cooperation with the nuclear transcription factor PRRX1 to regulate EMT, tumor cell migration and invasion, and dormancy in HNSCC [65]. Over-expression of PRRX1 leads to a cadherin switch with reduced E-cadherin and enhanced N-cadherin, and enhanced vimentin, SLUG, and ZEB2 levels (termed SIP1 in that publication). Inhibition of TGFβ signaling with SB431542 in cells over-expressing PRRX1 reversed the observed regulation of EMT-associated genes, migration, and invasion [65]. TGF-β1 also promotes STAT3 expression and enhanced binding to the promoter of the metastasis-associated lung adenocarcinoma transcript 1 (malat1). Malat1 in turn cooperates with miR-30a in inducing EMT and fostering the formation of HNSCC [57] (Figure 2A).

Puram and colleagues demonstrated the involvement of TGF-β1 derived from cancer-associated fibroblasts (CAFs) in pEMT induction in OSCC (see additional description in the chapter “EMT gene signatures”) [44]. In accordance with an induction of pEMT by CAFs that are resident in the surrounding tumor micro-environment (TME), the authors observed a preferential expression of pEMT markers (i.e., laminins B3 and C2 and podoplanin) at the interphase of malignant cells and the TME in primary OSCC [44]. We described a similar preferential loss of the epithelial marker EpCAM along with a gain of the mesenchymal marker Vimentin at the edges of tumor areas in HNSCC [66]. Comparably, while some primary tumors were characterized by a homogeneous expression of the EMT transcription factor (EMT-TF) SLUG, a preferential strong expression at the edges of tumor areas was associated with recurrences [67]. Spatially distinct induction of EMT and pEMT represents a source of ITH and plasticity that may support tumor bud formation, MRD, treatment resistance, and local/loco-regional invasion in HNSCC. Additionally to its role in tumor bud formation and in the metastatic cascade, TGF-β1 participates in the development of treatment resistance following therapeutic EGFR inhibition via an impairment of immune responses in cooperation with prostaglandin E2 [68]. In nasopharyngeal carcinomas (NPC), TGF-β1 functions can be modulated from metastasis enhancer to tumor suppressor functions by the transcription factor FOXA1, showcasing dual functions of EMT regulators as will be discussed for EGF, too [69]. Furthermore, regulation of TGF-β1 expression and activity by microRNAs (miRNAs) can be altered, for example, through polymorphisms in miRNA-binding sites, resulting in enhanced susceptibility towards HPV16-induced oropharyngeal cancer [70]. Additionally, a role for the TGF family member TGF-β2 in the regulation of dormancy of disseminated HNSCC tumor cells in the bone marrow was described, which was dependent on TGF-β receptors I and III [71]. TGF-β2-induced dormancy of disseminated HNSCC cells was shortened in lungs, resulting in metastatic outgrowth in this metastasis-permissive environment.

In addition to the direct effects of TGF-β1 signaling on HNSCC cells, TGF-β1 together with interleukin-17 (IL-17A) primes tumor-associated neutrophils to adopt a tumor-promoting phenotype and induce EMT-related functional changes in OSCC [72]. In SCC, IL-33-induced production of TGF-β1 by macrophages present in the niche of tumor-initiating cells (TICs) promotes invasion and drug-resistance in mouse models [73]. Comparable TGF-β1-dependent processes might be functional in HNSCC, too, and may impact the TME and immune cell composition [74].

Thus, TGF-β signaling is instrumental in the induction of EMT and of an invasive phenotype and contributes to tumor cell dormancy and treatment resistance in head and neck malignancies (Figure 2 and Figure 3).

#### 3.1.2. EGFR-Dependent EMT Regulation

A second important pathway that is pivotal in the regulation of cellular phenotypes and functions including EMT in HNSCC initiates at the EGFR. In the late 1980s and early 1990s, reports demonstrated that increased expression of EGFR was associated with malignant transformation and uncontrolled growth in HNSCC [4,75,76,77]. Induction of cell proliferation by the EGF/EGFR and the TGFα/EGFR axis in HNSCC [78,79] suggested a therapeutic potential for its inhibition using antagonistic monoclonal antibodies [80,81]. Chimeric mouse/human EGFR-specific monoclonal antibody Cetuximab was developed [82,83] and tested in combination with irradiation of HNSCC patients [23]. Cetuximab (tradename Erbitux in the US and Canada) received approval for combination treatment with irradiation in 2004 by the European Medicines Agency (EMA) and in 2006 by the Food and Drug Administration (FDA). Based on the EXTREME trial, Cetuximab received FDA clearance for first line treatment of locally advanced and/or recurrent/metastatic carcinomas in combination with platinum-based therapy plus fluorouracil owing to improved OS and PFS [84,85].

In addition to its function as a growth factor, EGF promotes cell migration, invasion, and metastasis formation through regulation of cell–cell adhesion. One molecular mediator of EGF-induced migration is focal adhesion kinase FAK [86], which is central to the dynamic formation and decomposition of focal adhesion points that are required for cell migration. EGF stimulation of breast carcinoma and oral squamous carcinoma (OSCC) cells that over-express EGFR promotes the inactivation of FAK through dephosphorylation [86]. FAK inhibition decreases cell–matrix adhesion and enhanced cell motility. Upon renewed adhesion of cells in an integrin-dependent manner, FAK function is restored, allowing for re-attachment of cells before the next cycle of detachment and re-attachment, thereby generating motility [86]. One mode of EGF-mediated disruption of cell–cell adhesion and induction of EMT in skin cancer, NSCLC, and prostate cancer cells relies on caveolin-dependent endocytosis of the major epithelial cell adhesion molecule E-cadherin. Subsequently, EMT-TF SNAIL (SNAI1) and ß-catenin-TCF/LEF-1 transcriptional functions are activated [87]. Similarly, EGF promotes cell migration and invasion, loss of E-cadherin and the adoption of a mesenchymal phenotype along with stem-like features in HNSCC [88,89,90,91,92]. EMT-related morphologic changes induced by constitutive EGFR signaling in HNSCC cell lines were associated with increased levels of SNAIL and reduced response to irradiation or Cetuximab monotherapy. However, EGF-mediated EMT enhanced the response to a dual treatment with a combination of irradiation and Cetuximab, suggesting a treatment-sensitizing function of Cetuximab and a potential window of therapeutic treatment [90]. Blocking of EGF-mediated EMT, cell migration and invasion, and metastasis formation by Cetuximab treatment was also demonstrated in OSCC, HNSCC, and esophageal squamous cell carcinomas [88,93,94,95].

With respect to the underlying molecular mechanisms, Xu and colleagues reported that EGF induces EMT and the acquisition of cancer stem cell traits via the EGFR/PI3K/HIF1α axis to enhance glycolysis in OSCC [91]. Accordingly, glycolysis inhibitor 2-deoxy-D-glucose (2-DG) reversed EGF-mediated EMT in vitro and reduced lymph node metastasis formation [91]. Gao et al. reported on the induction of EMT in HNSCC via the second major signaling branch of EGFR, i.e., MAPK/ERK1/2, which was triggered by EGF released by tumor-associated macrophages in the TME [96]. High levels of EGFR combined with low levels of epithelial cell adhesion molecules (EpCAM) correlated with the poorest overall and disease-free survival (OS and DFS), whereas the opposite constellation was associated with favorable survival of HNSCC [88]. At the molecular level, these clinical findings were explained by a dual ability of EGFR signaling to induce proliferation upon moderate activation and to promote EMT upon sustained, strong activation. ERK1/2 was identified as a major molecular switch involved in the decision-making of proliferation versus EMT. High levels of ERK1/2 phosphorylation were required for EGF-mediated EMT induction in cell lines and was associated with poorer OS in HNSCC patients [88]. In line with these findings, ERK2 over-expression is a common characteristic of recurrent HNSCC [97] (Figure 2A). A molecular basis for the observed interplay between EGFR and EpCAM was further reported by our group and the group of H.C. Wu in HNSCC and colon cancer, respectively [88,98]. A novel role for the soluble ectodomain of EpCAM termed EpEX, which is generated upon regulated intramembrane proteolysis by ADAM proteases [99], was determined in HNSCC [88], colon/colorectal cancer cells [98,100], and mesenchymal stem cells [101]. EpEX binds to the extracellular domain of EGFR as a functional ligand that induces classical signaling pathways engaged by EGFR including the MAPK and AKT pathways [88,98,100]. However, the strength of signaling induced by EpEX appeared inferior to high-dose EGF and did not result in the activation of EMT [88]. In HNSCC, co-treatment of cells with EMT-inducing concentrations of EGF and equimolar amounts of EpEX blocked EMT induction [88]. The exact mechanism underlying the capacity of EpEX to block EGF-induced EMT remains unclear so far.

Availability and spatial distribution of EGF within tumors will most probably further contribute to intra-tumor heterogeneity, as has been described for TGFβ during the induction of pEMT [44,102]. Multi-omics analyses of HPV-negative HNSCC showed that EGFR amplification, mRNA and protein levels, and activation levels of EGFR via phosphorylation were all congruent. However, these features of EGFR did not or only poorly correlate to the downstream pathway activity. The availability of EGFR ligands influenced pathway activation along the PI3K/AKT/mTOR and the RAF/MEK/ERK pathways, not the abundance of EGFR [21]. This important finding led to the conclusion that adjuvant treatment with antibodies targeting EGFR should be based on EGFR ligand and not EGFR abundance [21]. Thus, spatial differences regarding EGFR activation and repercussions on tumor cell differentiation are governed at various levels including ligand resources and represent a determinant of EMT heterogeneity in HNSCC.

Further regulation of the EGFR pathway to promote proliferation and EMT are conveyed by NOTCH1-inactivating mutations observed in >30% of HNSCC [6]. Mutated membrane-tethered variants that inactivate canonical NOTCH1 signaling can activate cell proliferation and EMT through the induction of the EGFR/PI3K/AKT axis [103]. Lastly, EGFR-dependent and -independent regulation of EMT was reported in HNSCC cells that initially depended on EGFR signaling. Tumor cells resistant to the EGFR inhibitor Erlotinib displayed enhanced Hedgehog signaling in the form of GLI1 expression and traits of EMT that were independent of EGFR [104]. Hedgehog signaling provided tumor cells with an escape mechanism from anti-EGFR treatment, which is dependent on EMT induction but not on EGFR signaling. Accordingly, a dual treatment with Cetuximab and the Hedgehog inhibitor IPI-926 in a xenograft mouse model was efficient in repressing tumor recurrence [104].

Hence, sustained EGFR activation is achieved through multiple molecular mechanisms and contributes to EMT-related intra- and inter-tumor heterogeneity in HNSCC. Insights into these functions may help to refine the selection of patients who are eligible for EGFR-based treatment modalities through the combinatorial assessment of EGFR ligand abundance, pathway activation, and potential gene signatures of EGFR-mediated EMT (Figure 2A and Figure 3).

#### 3.1.3. EMT Transcription Factors in HNSCC

Ultimately, EMT is regulated at the transcriptional level by six core transcription factors (EMT-TFs) SNAIL, SLUG, ZEB1 and 2, and TWIST 1 and 2 [49] (Figure 2A). EMT-TFs primarily act as transcriptional repressors with both redundant and non-redundant function that suppress the expression of epithelial genes mainly involved in cell–cell contact (Figure 2A). As a result, epithelial cells become disconnected, lose their polarity, and adopt a migratory phenotype that promotes tumor dissemination and metastasis formation [49,105] (Figure 2B). In addition to shared function in the induction of EMT, EMT-TF have non-redundant functions that affect a stem-like phenotype, cell metabolism and survival, and can differ depending on the cancer entity [105]. A recent meta-analysis of studies on the value of EMT-TFs for the prognosis of HNSCC demonstrated that SNAIL and SLUG (termed SNAI1 and 2), Twist 1, and Zeb1 are prognostic markers of poor OS [106]. Several reports have corroborated a central function of SLUG in EMT induction and a role as a prognostic marker associated with poor clinical outcome in HNSCC. SLUG is responsible for the switch from E- to N-cadherin under hypoxic conditions and following overexpression HIF-1α in HNSCC cell lines. In primary HNSCC, expression levels of SLUG and HIF-1α correlated and, together with a cadherin switch, defined patients with poor OS [107,108]. A SLUG-dependent cadherin switch in oral carcinoma cells was congruent with a re-organization of *adherens* junctions and loss of desmosomes [109]. Upon activation by the classical WNT/β-catenin signaling pathway, SLUG provides cells with increased mobility and supports the invasive phenotype of HNSCC cells required for lymph node metastases formation [110]. In line with a strong influence of lymph node metastases on the prognosis of HNSCC patients, high expression of β-catenin and SLUG was associated with the presence of nodal metastases and reduced survival [110]. High levels of phosphorylated ERK1/2 and/or SLUG correlated with decreased OS [88] and digital scoring of SLUG protein expression was an independent prognostic marker of recurrence-free and disease-specific survival [111]. SLUG induces the expression of stem cell markers, enhanced resistance to cisplatin, and was associated with tumor cell invasion in HNSCC xenografts in the mouse [112]. Furthermore, SLUG was associated with the expression of a 15-gene signature of partial EMT defined in oral carcinomas, which was correlated with the nodal status of patients (see “EMT gene signatures”) [44,67]. However, enhanced SLUG expression correlated with the pEMT signature in tumors of individual patients, but this correlation was not observed at the single cell level in primary tumors [44]. High expression of SLUG in primary HNSCC was a marker for reduced disease-free survival and was linked to tumor recurrence and the expression of EGFR [67]. Based on the ability of SLUG to promote treatment resistance, Riechelmann and colleagues compared the clinical outcome of HNSCC patients receiving either primary radiotherapy (RT) or radiochemotherapy (RCT) with patients receiving surgery with or without subsequent adjuvant RT/CRT in dependency of SLUG expression. Patients with SLUG-expressing HNSCC had a substantial benefit from a treatment regimen comprising an upfront surgery, which fortified a role for SLUG in resistance towards RT and RCT [113]. Unfortunately, SLUG is not considered a druggable target, but regulators and/or co-factors of SLUG may serve as surrogate targets. Since ERK1/2 is a major switch between proliferation and EMT induction in HNSCC [88] and ERK2 is upregulated in recurrent tumors [97], a specific targeting of ERK signaling may represent a druggable option.

EMT-TF SNAIL is an additional driver of EMT in HNSCC that suppresses E-cadherin-mediated cell–cell adhesion and supports anchorage-independent growth and resistance to the EGFR inhibitor erlotinib and cisplatin [114,115]. SNAIL is instrumental in p38 MAP kinase-mediated and in IL6-dependent EMT in HNSCC [116,117]. Similarly to SLUG, SNAIL promotes EMT along with the development of a stem-like phenotype [115,118]. Somewhat counterintuitive to the role of SNAIL in EMT and suppression of cell-cell contact, Li et al. have demonstrated that SNAIL induces the expression of the tight junction protein claudin-11. Upon activation through tyrosine phosphorylation, claudin-11 activates Src kinase to inhibit RhoA and thereby fosters cell–cell adhesion. As a result of this process, SNAIL does not only induce EMT but also the collective migration and invasion of tumor cells and the formation of clusters of circulating tumor cells in HNSCC patients, which correlated with nodal metastases, recurrence, and poor outcome [119].

Reports on TWIST functions HNSCC are scarcer but support a role in β-catenin- and AKT-mediated EMT in HNSCC [120] and identified TWIST1 and 2 as prognostic markers in OSCC and HNSCC [121,122].

### 3.2. EMT Gene Signatures

One basic assumption is that functional traits associated with EMT deteriorate the outcome of cancer patients and, therefore, that the degree of EMT correlates with clinical outcome. Hence, the identification of pEMT gene signatures and surrogate biomarkers to improve patient stratification, prognosticate clinical outcome, and offer therapeutic options have gained considerable interest. In 2006, Chung et al. reported the identification of a 75-gene list that was extracted using DNA microarrays, which defined HNSCC patients at high risk of recurrence. Gene set enrichment analysis (GSEA) revealed eight significantly enriched gene sets in high-risk patients, four out of which were related to EMT, NF-kB signal transduction, and cell adhesion [123]. Specific genes in EMT gene sets included matrix metalloproteinases (MMP-2 and MMP-12), which may foster the tissue-remodeling capacity of tumor cells during local and distant invasion. MMP2 induction in HNSCC was also observed in patients over-expressing the Nijmegen breakage syndrome 1 (NBS1) gene along with an up-regulation of the EMT-TF SNAIL and worsened clinical outcome [124]. Chemoattractants such as CXCL1 may help in recruiting non-malignant supportive cells such as neutrophils, which have ample functions as tumor-associated neutrophils (TANs) in the tumor environment [125]. Furthermore, platelet-derived growth factor receptors alpha and beta (PDGFR-A/B) have been associated with an induction of resistance to cisplatin, anti-apoptotic traits, and metastases formation [126,127]. Four out of eight gene sets enriched in low-risk patients were associated with mRNA metabolism and cell cycle regulation [123], which could potentially explain a better response to R(C)T in cycling cells and subsequent reduction of recurrences.

Puram et al. performed a single-cell RNAseq (scRNAseq) analysis of *n* = 18 patients with OSCC generating more than 6,000 single cell transcriptomes of malignant and non-malignant cells. Non-malignant stromal and immune cells did not significantly differ in their transcriptional programs across all patients whereas malignant cells showed intra- and intertumor variations [44,102]. These different transcriptional programs were categorized as cell cycle, hypoxia, stress, epithelial differentiation, and pEMT, the latter program being associated with the presence of nodal metastases [44,102]. The pEMT signature was defined as *n* = 15 common pEMT genes from a total set of *n* = 100 genes. A subsequent immunohistochemical analysis of three pEMT markers (Podoplanin, LAMB3, and LAMC2) in conjunction with one epithelial marker (SPRR1B) demonstrated a correlation of the higher expression of pEMT markers with worse differentiation, nodal involvement, and perineural invasion but no significant association with OS and DFS in OSCC [128]. A prognostic value of pEMT or EMT markers has been validated in clinical cohorts of HSNCC and OSCC [129,130,131].

ScRNAseq has the considerable advantage of delivering information on the heterogeneity and the contribution of EMT at the single cell level. However, scRNAseq remains technically demanding and has not entered clinical routine. Therefore, we aimed at transferring findings from scRNAseq to larger cohorts of HNSCC and correlating pEMT with clinical endpoints. To do so, bulk RNAseq and micro-array results from The Cancer Genome Atlas program (TCGA), the MD Anderson Cancer Center (MDACC), and the Fred Hutchinson Cancer Center (FHCC) were deconvoluted using the EPIC algorithm [132] and, where required, were cleared for patients with high content of CAF in TCGA. CAF are cells of mesenchymal origin that showed the strongest correlation level with the pEMT signature amongst non-malignant cell types. The resulting large cohorts served to compute a risk score based on the expression of the 15 common pEMT genes using single-sample scoring of molecular phenotypes (Singscores) [133]. pEMT-Singscores were identified as an independent prognosticator of OS in all three cohorts and as an independent prognosticator of OS in the sub-cohort of irradiated patients of TCGA [67]. An analysis of differentially expressed genes and related GO-terms showed that pEMT is associated with an up-regulation of cell motility and a down-regulation of epithelial differentiation and oxidative phosphorylation [67]. Correlation analyses further defined SLUG as most strongly associated with single genes of the pEMT signature and with pEMT-Sinscores in patients and cell lines. Stable expression of SLUG in cell lines partly recapitulated pEMT features including cell invasion and enhanced resistance to irradiation. Accordingly, strong, and preferentially peripheral expression of SLUG protein was associated with recurrence and reduced PFS in an additional HNSCC cohort [67].

In their recent publication, Tyler and Tirosh further tackled the issue of EMT signatures in cancer that arises from the presence of stromal cells of mesenchymal origin such as CAFs. The authors have compared EMT signatures from scRNAseq of tumor and stromal cells and achieved a discrimination of both signature types in bulk RNAseq datasets [134]. Screening a large variety of cancers, EMT was not correlated with enhanced metastasis formation, which led the authors to hypothesize that “other steps in the metastatic cascade may represent the main bottleneck” [134]. Interestingly though, HNSCC stood out because of a correlation of pEMT with the presence of lymph node metastases and the N-status, again leading the authors to state that “the extent of pEMT may be particularly important for metastasis and N-stage in the classical subtype of HNSCC” [134]. A correlation of pEMT with resistance to therapy was further highlighted for HNSCC of the malignant basal subtype, stressing the idea that EMT and pEMT are implicated in multiple processes of HNSCC progression.

In conclusion, pEMT quantification represents a valid molecular prognosticator to be further developed in retrospective and, more importantly, in prospective HNSCC cohorts.

### 3.3. EMT, Stem-Like Properties, and Treatment Resistance

As already mentioned, the role of EMT in cancer progression is not restricted to the induction of an invasive phenotype. EMT has been frequently associated with a (cancer) stem-like phenotype and enhanced resistance to standard therapy. In HNSCC cell lines, sphere-forming cells showed an enriched expression of the cancer stem cell (CSC) markers ALDH-1 and CD44 and the pluripotency genes Sox2, Oct3/4, and Nanog. These CSC-like cells were more invasive and were characterized by EMT features (expression of α-SMA and vimentin, and reduction of E-cadherin) [135]. Two distinct types of CSC that are either in an epithelial state (CD44^high^/EpCAM^high^) or in a mesenchymal state (CD44^high^/EpCAM^low^) have been described, which are rather proliferative or migratory, respectively [136,137]. Furthermore, GSK3β and TRAF6 were decisively involved in the maintenance of a CSC phenotype and the induction of EMT in HNSCC cells [138,139]. EMT-related resistance to the EGFR inhibitor Erlotinib can be induced via nerve growth factor (NGF) and tropomyosin receptor kinase A TrkA), which are both elevated in HNSCC and predict poor survival [140]. Furthermore, EMT enhanced resistance to irradiation in vitro [141], was induced by the polycomb group protein enhancer of zeste 2 (EZH2) and was associated with reduced sensitivity towards cisplatin in patients and in cellular models [142]. Several other reports linked EMT induced by EGFR, Akt, and other signaling pathways to enhanced resistance to various treatments [143,144,145,146,147]. However, the precise connection and the underlying mechanisms of stemness, EMT, and treatment resistance in HNSCC remain incompletely understood.

## 4. Consequences for Treatment

Depending on UICC-based stratification and standard treatment regimens in place at hospitals around the world, HNSCC patients will either receive a multi-modal therapy (Figure 3A) or definitive RCT (Figure 3B). Multi-modal therapy is comprised of a surgical resection of the primary tumor and loco-regional lymph nodes during a neck dissection. Depending on clinical staging, RT or R(C)T can be applied as adjuvant treatments to improve disease control. Definitive R(C)T is based on RT or RCT without prior surgical removal of the primary tumor. Definitive R(C)T may select for treatment-resistant clones that can provoke local recurrences and nodal metastases. Multi-modal treatment on the other hand will more likely face the problem of tumor cells that have already detached from the primary tumor and have either formed a local MRD or have disseminated at the time point of surgery. Cancer treatment in the form of definitive or adjuvant R(C)T and various therapeutic biologicals can act as potent selectors of therapy-resistant cells, leading to the extinction of sensitive cancer cell clones while potentially promoting resistant ones [148]. Several mechanisms of treatment resistance may come into effect before treatment initiation, including (p)EMT [149,150,151]. pEMT enables cancer cells to invade surrounding tissue and migrate away from the tumor, and, thus, increases the risk of MRD after tumor resection and lymph node dissection, thus potentially impeding on multi-modal therapy (Figure 3A). On the other hand, pEMT increases the intrinsic resistance to radio(chemo)therapy and can impact definitive R(C)T (Figure 3B). Thus, (p)EMT may impact multi-modal and definitive RCT treatment schemes of HNSCC at different stages of the disease. It should further be noted that HNSCC patients mostly suffer from local recurrence and/or local lymph node metastases that arise through lymphatic spread. Distant metastases that are the result of hematogenous spread remain infrequent in HNSCC. Although still speculative in nature, it is conceivable that (p)EMT plays a more central role in HNSCC than other carcinomas as drivers of treatment resistance, local and lymphatic spread, as recently postulated by Tyler and Tirosh [134].

Despite the negative impact on disease control, (p)EMT may likewise represent a novel opportunity to improve stratification, patient care, and potentially therapeutic approaches [152,153]. However, clinical trials specifically addressing EMT in HNSCC are scarce with currently two entries in ClinialTrials.gov with the search terms “HNSCC” and “EMT” (NCT01927354 and NCT02119559). Monoclonal therapeutic antibodies specific for TGFβR are in early clinical phases (phase I and II) to test potential benefits for patients in advanced stages of carcinomas, including esophageal carcinomas [154]. However, patient numbers remained small and only two patients with ESCC were treated in this published study [154]. The maximal tolerable dose (MTD) could not be determined because dose escalation was not considered safe owing to adverse effects in the form of a cytokine storm. A similar outcome was reported for the TGFβR kinase inhibitor YL-13027, including the inability to define an MTD (Clinical trial identifier NCT03869632). In 2020, an open label trial testing the bifunctional fusion protein SHR-1701 that targets PD-L1 and TGFβ has been launched with *n* = 48 patients suffering from non-small cell lung cancers with EGFR mutations (Clinical trial identifier NCT04324814), but no results are available thus far. Further observational studies on the regulation of EMT in HNSCC patients have been conducted or are ongoing. For example, the role of miRNAs in regulating EMT-TFs in HNSCC patients was investigated, but study results have not been disclosed so far (Clinical trial identifier NCT01927354). Since targeting EMT-TFs is very challenging, alternative strategies have been proposed. Repurposing inhibitors of metabolic pathways for the treatment of EMT is such an option that could be combined with standard regimens [155]. Inhibitors of the EGFR pathway are in late clinical phases (III and IV) or have approval for the treatment of HNSCC (e.g., Cetuximab). However, to the best of our knowledge, the application of EGFR inhibitors with the aim to therapeutically address EMT in HNSCC remains poorly studied. Most studies involving EGFR inhibitors are of preclinical nature except for the histone deacetylase (HDAC) inhibitor valproic acid. Caponigro et al. describe the launch of a phase II study implementing valproic acid together with cisplatin and cetuximab for the treatment of advanced HNSCC within the V-CHANCE trial [156]. The study has finalized patient recruitment and the estimated study completion date is February 2022. Preliminary results on potential benefits of valproic acid have not been made public.

Hence, clinical studies aiming at targeting EMT in HNSCC are scarce, in early stages, and to the best of our knowledge not based on EMT-related stratification of the enrolled patients. Next-generation RNA sequencing of biopsy material of primary HNSCC at first diagnosis could allow forming a (p)EMT-based risk score that can be further linked to various signaling pathways and downstream gene activation via automated bioinformatic pipelines. Signaling pathways strongly correlated to (p)EMT signatures in patients may uncover personalized therapeutic options in the form of available inhibitors of receptors and pathways (e.g., TGFβR, EGFR, Hedgehog, WNT, etc.) for multi-modal and definitive R(C)T regimens alike. Such inhibitors may be combined with R(C)T in the primary or adjuvant setting, and/or may serve as second/third line treatment in the case of disease progression in a highly targeted and personalized manner (Figure 3). Unlike the current situation in which inhibitors may be administered even in the absence of knowledge of the actual presence of the therapeutic molecule on target cells, such an approach would foster a considerably more targeted application of already approved inhibitors and could possibly enhance their efficacy. Additionally, (p)EMT risk scores generated from primary tumors may help to adjust R(C)T regimens with respect to dose and irradiation field in multi-modal treatment and definitive R(C)T.

Therefore, a better understanding of how cancer cells exploit programs such as pEMT and of initiating extracellular clues, signaling pathways, transcription factors, and epigenetic regulation seems a promising approach to find new therapeutic targets of p16-negative HNSCC and improve the treatment of this highly heterogeneous and impressively malignant disease.

## 5. Conclusions

The evaluation of genetic alterations regarding their functional impact on cancer cells remains a huge task in cancer biology. Despite that, reasonably well-characterized physiological programs such as hypoxic response, multidrug resistance, and EMT were shown to be exploited by cancer cells, presumably independent from their genetic background. The transcriptomic activation of these epigenetically regulated programs crucially contributes to tumor cell plasticity and therapy resistance. Multiple lines of evidence identify (p)EMT as a major contributor to ITH and treatment failure in HNSCC. Thus, a deeper insight into the involved molecular pathways seems to promise a timely approach of intervention.

## Figures and Tables

**Figure 1 cancers-13-05355-f001:**
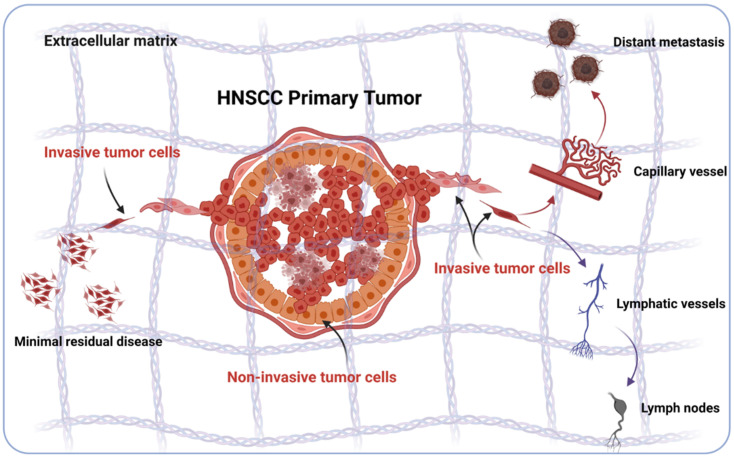
Schematic representation of the involvement of EMT in local, loco-regional, and distant invasion in HNSCC. Sub-populations of HNSCC cells in various states of EMT can loosen cell–cell contacts, detach from the primary tumor, and locally invade surrounding tissue. Locally invading tumor cells can remain as a minimal residual disease despite surgery and resist multi-modal therapy. Alternatively, locally disseminated tumor cells can gain access to lymphatic and blood vessels, colonize loco-regional lymph nodes or distant organs and form metastases (Figure generated using BioRender, Toronto, ON, Canada).

**Figure 2 cancers-13-05355-f002:**
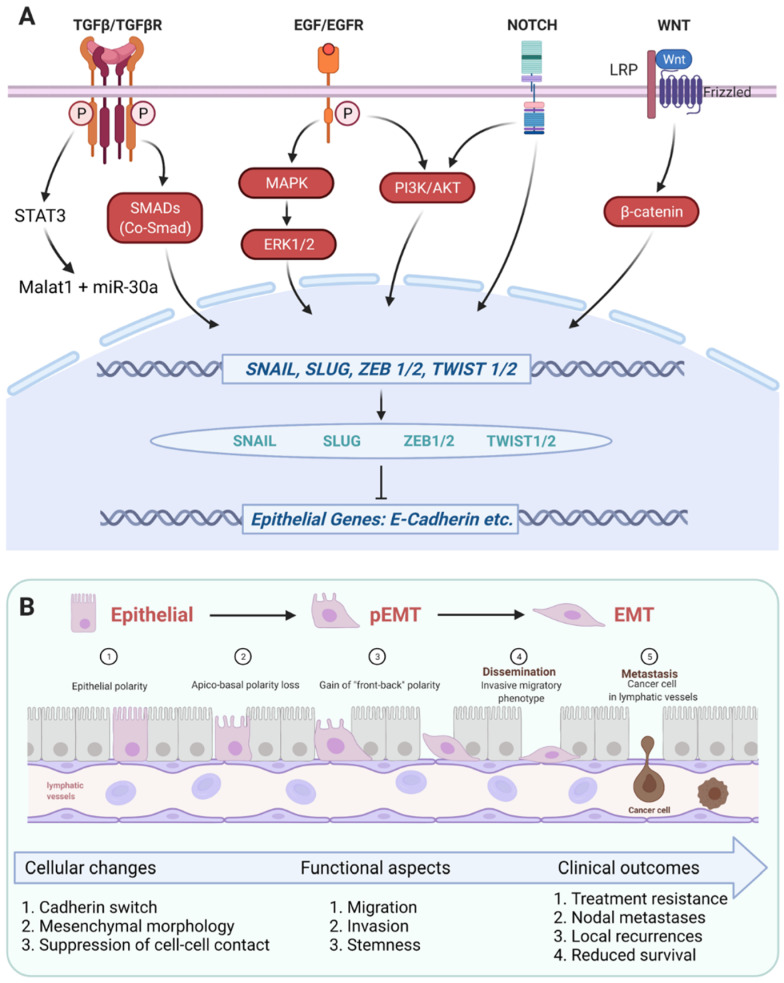
Schematic representation of central signaling pathways involved in the induction of (partial) EMT in HNSCC. (**A**) Figure 2A depicts major ligands, receptors, signaling components (red color) and transcription factors involved in the induction of EMT and partial EMT (genes are depicted in blue color, proteins in green color) via the repression of epithelial genes (depicted in blue color). TGFβR, EGFR, NOTCH, and LRP/Frizzled receptors have been reported to induce the expression of EMT-TFs SNAIL, SLUG, ZEB1/2, and TWIST1/2 through the binding of their cognate ligands, i.e., TGFβ, EGF, Delta/Jagged or mutations, and WNT variants, respectively. TGFβR signaling towards EMT is functional via signal transducer and activator of transcription 3 (STAT3) and the subsequent induction of Malat1 and miR-30a, and via SMADs/co-SMAD. Activation of EMT through EGF/EGFR was reported to depend primarily on MAPK and ERK1/2; however, induction via PI3K and AKT was described too. Uncleavable mutated variants of NOTCH receptors were shown to trans-activate EGFR signaling at the level of PI3K and AKT, converging in EMT induction. WNT signaling via the Frizzled/LRP receptors result in nuclear translocation of β-catenin. (**B**) Figure 2B depicts the cellular, functional, and clinical consequences of the induction of (p)EMT described in Figure 2A. Molecular and cellular changes associated with (p)EMT include a progressive transit from an epithelial polarity (①) to a loss of apico-basal polarity (②) and a gain of “front-back polarity” (③), a switch in cadherin variants expression from E-cadherin to N-cadherin, the adoption of a mesenchymal morphology due to a suppression of cell–cell contacts. As a result, cells in EMT are characterized by enhanced migration, invasion, and stem-like properties (④) that influence treatment resistance, metastases formation (⑤), and recurrences, and thereby affect the clinical outcome of HNSCC patients (Figure generated using BioRender, Toronto, ON, Canada).

**Figure 3 cancers-13-05355-f003:**
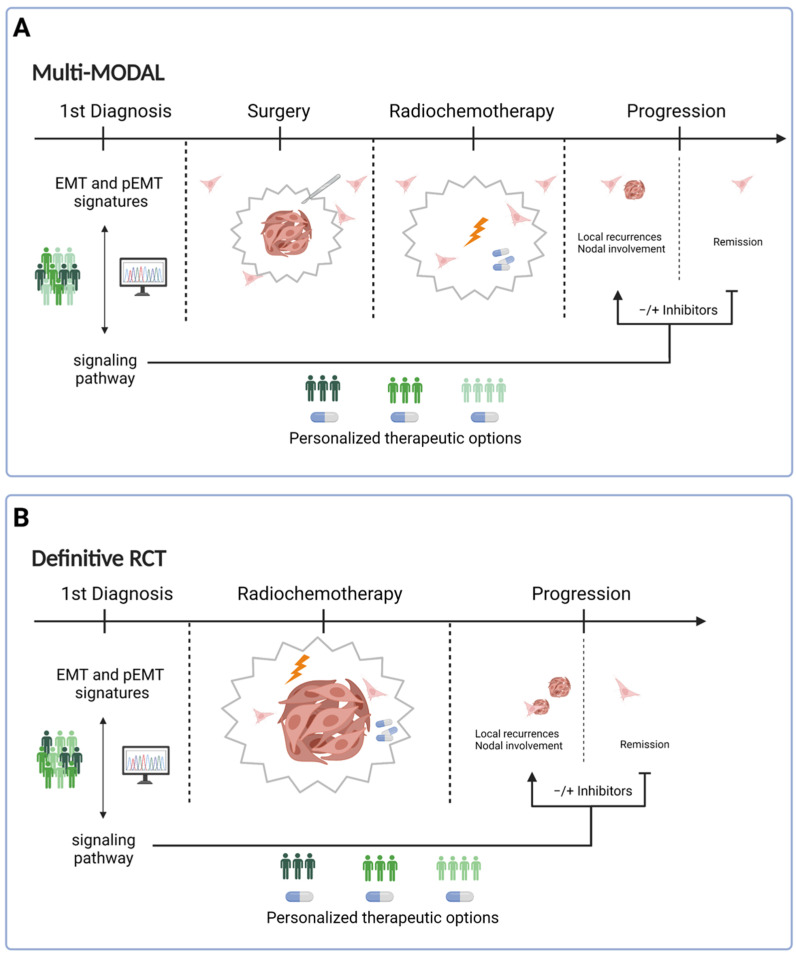
Potential implementation of (p)EMT signatures in the treatment of HNSCC. Two major treatment modalities can be defined as (**A**) multi-modal therapy and definitive radio(chemo)therapy (**B**); definitive R(C)T). Multi-modal therapy includes a surgical resection of the primary tumor and the loco-regional lymph nodes, and adjuvant R(C)T. Definitive R(C)T relies on radiotherapy or radio- and chemotherapy without prior surgical removal of the primary tumor. In both treatment modalities, RNA sequencing of primary HNSCC from biopsy material at initial diagnosis can help to develop (p)EMT risk scores based on validated EMT signatures [44,67]. Bioinformatic pipelines can identify potential correlations of (p)EMT risk scores with signaling pathways that are targets for approved inhibitors, which can thereby be administered in a personalized manner in multi-modal (**A**) and definitive R(C)T treatment regimens (**B**). (Figure generated using BioRender, Toronto, ON, Canada).

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
