# Peer review of "Epithelial-to-Mesenchymal Transition-Derived Heterogeneity in Head and Neck Squamous Cell Carcinomas"

_cancers, 2021, doi:10.3390/cancers13215355_

Round 1
Reviewer 1 Report
the subject of the Review seems very interesting and that there are not many published reviews on this subject.
However, the first thing I have noticed is that they write in a very complex way on many occasions. It is difficult to read and understand in certain paragraphs, they should edit the text to simpler sentences.
On the other hand, all the time they take for granted that the phenomenon of TMS is directly related to stemness and resistance to treatment, and even that TMS activates the stemness pathways ("Cancer stemness features are fostered via EMT. .. ", among many other examples). It gives some contradictory feeling that at the end they say this: "However, the precise connection and the underlying mechanisms of stemness, EMT, and treatment resistance in HNSCC remain incompletely understood." They should speak of TMS as a process that initiates what will later be metastasis to distant nodules or organs, but not as the basis of all resistance to treatment and stemness. I think that resistance to treatment can occur without TMS and vice versa, although it may be related in terms of signaling pathways or in other ways, as occurs with the acquisition of the characteristics of stemness. This should be well explained in the text.
Regarding the signaling pathways that they speak of, they are very general and ubiquitous pathways, which although they are preferentially activated in cancer and head and neck cancer, I think that they not only activate the EMT as they continually point out, but also they will participate in stemness, proliferation, etc. There are paragraphs that seem contradictory:
- "EMT-related morphologic changes induced by constitutive EGFR signaling in HNSCC cell lines were associated with increased levels of SNAIL and reduced response to irradiation and Cetuximab treatment. However, EGF-mediated EMT enhanced the response to a combination of irradiation and Cetuximab treatment, suggesting a potential window of therapeutic treatment. Blocking of EGF-mediated EMT, cell migration and invasion, and metastasis formation by Cetuximab treatment was also demonstrated in OSCC, HNSCC, and esophageal squamous cell carcinomas. "
- "EpEX binds to the extracellular domain of EGFR as a functional ligand that induces classical signaling pathways engaged by EGFR including the MAPK and AKT pathways. In HNSCC, co-treatment of cells with EMT-inducing concentrations of EGF and equimolar amounts of EpEX blocked EMT induction. "
- "SLUG correlated with the pEMT signature at the individual tumor level but not at the single tumor cell level." ? Finally, when it comes to making a gene signature or trying to stratify patients according to the expression of certain genes that indicate TMS, I think it would be very complicated and even more so in head and neck cancer that is so heterogeneous, but good we always have that problem.
Also, in some parts of the article, the information seems a bit messy. Some more specific corrections are the following ones:
Line 22: change HNSC to HNSCC.
Line 58: It seems to say that tobacco (and alcohol) is the sole cause of "field cancerization", but there may be other causes such as gastroesophageal reflux in the case of Barret's esophagus, so it should be indicated as an example of the cause.
Figure 1: As I see it, this image seems to be very general for any solid tumor, it is nothing specific to HNSCC. Anyway, it is fine.
Figure 2: 4 à change mirgratory to migratory.
Figure 2: The figure caption describes the images in a brief and correct way. However, in the upper schematic, there are details not mentioned in the figure caption that perhaps should be included (STAT3, Malat 1 + miR30a, MAPK, ERK1 / 2, PIK3 / AKT, β-catenin). In addition, a more complete scheme would have seemed interesting to me, which could help to follow the content of the article.
Line 156-157: “no specific mutations were reported to initiate or to be required to orchestrate this reversible transition to more mesenchymal features” à I´m not sure this sentence to be completely true. Maybe there is no confirmed mutations to diagnose, but some mutations have been suggested (AS:
Line 256: It seems that some connection link is missing here with the previous sentence, or perhaps a linking word such as "on the other hand", giving the text a better cohesion.
Line 309-330: From my point of view, this paragraph seems to be jumping between the different molecular mechanisms it describes. It lacks a little more order.
Line 317-318: the meaning of the acronyms is missing: DFS (disease-free survival) and OS (overall survival)
Line 504: Missing connectors or more cohesion in the text.
In lines 151-153 talk about three miRNAs, what are? More details.
It is neccesary to rewrite from line 196-199 because is not understood.
In lines 228-230 is neccesary to explain how TGFb1 cooperate with PRRX1 to regulate EMT.
The two sections 3.1.1 and 3.1.2 are a little confused without order and it is very difficult to follow the reading.
It seems to be neccesary to include a table with possible drugs or combination related EMT to reduce resistance and improve the individualized therapiesin HNSCC.
Author Response
Reviewer 1
the subject of the Review seems very interesting and that there are not many published reviews on this subject.
However, the first thing I have noticed is that they write in a very complex way on many occasions. It is difficult to read and understand in certain paragraphs, they should edit the text to simpler sentences.
Answer: We thank Reviewer #1 for pinpointing at an apparently utterly poor quality of writing that was rated in the lower 20% in the report file. Where possible, simpler sentences have been incorporated in the revised manuscript. We hope it helps to ease the reading of the manuscript.
On the other hand, all the time they take for granted that the phenomenon of TMS is directly related to stemness and resistance to treatment, and even that TMS activates the stemness pathways ("Cancer stemness features are fostered via EMT. .. ", among many other examples). It gives some contradictory feeling that at the end they say this: "However, the precise connection and the underlying mechanisms of stemness, EMT, and treatment resistance in HNSCC remain incompletely understood." They should speak of TMS as a process that initiates what will later be metastasis to distant nodules or organs, but not as the basis of all resistance to treatment and stemness. I think that resistance to treatment can occur without TMS and vice versa, although it may be related in terms of signaling pathways or in other ways, as occurs with the acquisition of the characteristics of stemness. This should be well explained in the text.
Answer: We are not aware of the abbreviation TMS and have not found any mention of it in relation to the topic of our review. Therefore, we have addressed Reviewer 1´s comment assuming the acronym TMS refers to EMT.
In response to the broader criticism regarding EMT and its relation to stemness and resistance to treatment: We do not believe that EMT is indispensable for stemness and treatment resistance, but that it can significantly contribute to both. However, exact mechanisms of EMT-mediated stemness and treatment resistance are not fully resolved, which was our rational to describe it as such. Further, we do believe that EMT or partial forms of it play an important role in the development of nodal metastases in HNSCC, which has been impressively demonstrated by Tyler and Tirosh in their recent Nature Communications publication [1]. We have aimed at improving the description of these aspects of EMT in our revision and hope it is satisfactory to the reviewer.
Regarding the signaling pathways that they speak of, they are very general and ubiquitous pathways, which although they are preferentially activated in cancer and head and neck cancer, I think that they not only activate the EMT as they continually point out, but also they will participate in stemness, proliferation, etc. There are paragraphs that seem contradictory:
- "EMT-related morphologic changes induced by constitutive EGFR signaling in HNSCC cell lines were associated with increased levels of SNAIL and reduced response to irradiation and Cetuximab treatment. However, EGF-mediated EMT enhanced the response to a combination of irradiation and Cetuximab treatment, suggesting a potential window of therapeutic treatment. Blocking of EGF-mediated EMT, cell migration and invasion, and metastasis formation by Cetuximab treatment was also demonstrated in OSCC, HNSCC, and esophageal squamous cell carcinomas. "
- "EpEX binds to the extracellular domain of EGFR as a functional ligand that induces classical signaling pathways engaged by EGFR including the MAPK and AKT pathways. In HNSCC, co-treatment of cells with EMT-inducing concentrations of EGF and equimolar amounts of EpEX blocked EMT induction. "
- "SLUG correlated with the pEMT signature at the individual tumor level but not at the single tumor cell level." ? Finally, when it comes to making a gene signature or trying to stratify patients according to the expression of certain genes that indicate TMS, I think it would be very complicated and even more so in head and neck cancer that is so heterogeneous, but good we always have that problem.
Answer: The present review is on EMT in HNSCC, which is why we have aimed at highlighting the different signaling pathways reported to induce EMT in HNSCC. Obviously, these pathways also have functions in non-pathologic and non-malignant conditions as well as in non-EMT-related processes such as proliferation and stemness. Given the already voluminous review we have refrained from addressing these additional topics in detail. Nonetheless, taking EGFR signaling as an example, we have clearly defined that in HNSCC cell lines, moderate activation results in proliferation whereas strong and sustained activation induces EMT. The notion that we continually point out that these pathways only activate EMT therefore appears a far stretch in opinion. We hope this helps to resolve potential misunderstandings.
More specifically to each bullet point:
- This sentence has been changed to clarify that EMT reduced the response of cells to mono-therapies with irradiation or Cetuximab, but that a combined treatment with irradiation and Cetuximab was effective.
- EpEX induces EGFR signaling but to an inferior level than high-dose EGF and does not induce EMT in vitro. This point has been clarified in the revised manuscript.
- We have clarified the sentence to explain that SLUG levels and the pEMT signature correlated in individual patients but not at the single cell level. This means that a patient with a high pEMT signature commonly also expressed high levels of SLUG. However, when using scRNAseq, SLUG expression and the pEMT signature level were not correlated within single cells. We agree that the definition of a clinically valuable signature of EMT is a difficult task, even more so when addressing the predictive potential of such a signature. However, attempts by the Tirosh, Regev, and Bernstein labs and ours have shown promising results in defining and transferring signatures established by high-resolution scRNAseq to bulk sequencing data of large clinical cohorts [1-3].
Also, in some parts of the article, the information seems a bit messy. Some more specific corrections are the following ones:
Line 22: change HNSC to HNSCC.
Answer: HNSCC has been changed accordingly.
Line 58: It seems to say that tobacco (and alcohol) is the sole cause of "field cancerization", but there may be other causes such as gastroesophageal reflux in the case of Barret's esophagus, so it should be indicated as an example of the cause.
Answer: The mention of reflux as a cause for field cancerization in the specific situation of Barret´s esophagus has been added.
Figure 1: As I see it, this image seems to be very general for any solid tumor, it is nothing specific to HNSCC. Anyway, it is fine.
Answer: Based on the comment, no changes have been made.
Figure 2: 4 à change mirgratory to migratory.
Answer: Figure 2 has been changed accordingly.
Figure 2: The figure caption describes the images in a brief and correct way. However, in the upper schematic, there are details not mentioned in the figure caption that perhaps should be included (STAT3, Malat 1 + miR30a, MAPK, ERK1 / 2, PIK3 / AKT, β-catenin). In addition, a more complete scheme would have seemed interesting to me, which could help to follow the content of the article.
Answer: The legend to Figure 2 has been changed to describe the content more comprehensively. Since Figure 2 concentrates on signaling pathways involved in EMT induction in HNSCC that are described in the manuscript, we refrained to add information to the Figure and make it even more complicated.
Line 156-157: “no specific mutations were reported to initiate or to be required to orchestrate this reversible transition to more mesenchymal features” à I´m not sure this sentence to be completely true. Maybe there is no confirmed mutations to diagnose, but some mutations have been suggested (AS:
Answer: This claim has been toned down and now reads “Currently available data does not suggest a major contribution of specific mutations in initiating or orchestrating this reversible transition to more mesenchymal features”.
Line 256: It seems that some connection link is missing here with the previous sentence, or perhaps a linking word such as "on the other hand", giving the text a better cohesion.
Answer: The word “Additionally” has been added to the sentence.
Line 309-330: From my point of view, this paragraph seems to be jumping between the different molecular mechanisms it describes. It lacks a little more order.
Answer: In this paragraph, we describe different ways used by EGFR to induce EMT in HNSCC. We have rephrased some of the sentences and hope it is now more structured.
Line 317-318: the meaning of the acronyms is missing: DFS (disease-free survival) and OS (overall survival)
Answer: both acronyms have been defined in the revised manuscript.
Line 504: Missing connectors or more cohesion in the text.
Answer: The word “Furthermore” was added as a connecting link.
In lines 151-153 talk about three miRNAs, what are? More details.
Answer: The three miRNAs have been named and it is now mentioned that they had not been linked to cancer previously.
It is neccesary to rewrite from line 196-199 because is not understood.
Answer: This passage has been re-written.
In lines 228-230 is neccesary to explain how TGFb1 cooperate with PRRX1 to regulate EMT.
Answer: PRRX1 effects on EMT were reversed upon inhibition of TGFβ signaling. This point was described in more detail in the revision.
The two sections 3.1.1 and 3.1.2 are a little confused without order and it is very difficult to follow the reading.
Answer: Chapter 3 describes EMT in HNSCC. Sub-chapter 3.1. addresses the regulation of EMT in HNSCC and from there on the various receptor-mediated signaling pathways, as is already mentioned in the original manuscript. With all due respect, we do not see how to improve this structure. We have opted to keep this part of the manuscript unchanged.
It seems to be neccesary to include a table with possible drugs or combination related EMT to reduce resistance and improve the individualized therapiesin HNSCC.
Answer: This is an excellent idea but unfortunately such a table would be highly speculative since clinical trials addressing EMT in HNSCC are very scarce (two entries in ClinicalTrials.gov). To the best of our knowledge, no prospective or retrospective trials specifically testing inhibitors towards regulators of EMT are available to date. To address this point of criticism nonetheless, we have opted to describe relevant trials in the chapter “4. Consequences for treatment” rather than as a table. We hope this is satisfactory to Reviewer 1.
Reviewer 2 Report
This is a reviewing report of EMT and pEMT in HNSCC, which is well organized, well summarized previously published manuscripts and stated about clinical therapeutic strategies. I think this review would definitely provide big interest to lots of readers not only for clinicians but also for basic researchers.
Some modifications would be required for easier understandings especially for figures 2 and 3. There is nothing to be modified for the text part, however, figure legends for figures 2 and 3 are not kind for beginners.
A more detailed explanation would be required including meaning of colour difference - red or brown for smads, MAPK, ERK1/2, PI3K/AKT and b-catenin, and red, brown, green, blue for snail, slug, zeb1/2, twist 1/2, and E-cadherin-. Moreover, each upper and lower panel appears to show different concepts for molecular aspect and translational of clinical and molecular aspect, respectively; however, there is not a direct explanation in figure legend (Figure 2).
For figure 3, a direct explanation for each panel would be favourable for better understandings for readers. The colour difference for "personalized therapeutic options" - red, green and blue- might be quite confusing considering each colour has different images in various countries and cultures.
Author Response
Reviewer 2
This is a reviewing report of EMT and pEMT in HNSCC, which is well organized, well summarized previously published manuscripts and stated about clinical therapeutic strategies. I think this review would definitely provide big interest to lots of readers not only for clinicians but also for basic researchers.
Some modifications would be required for easier understandings especially for figures 2 and 3. There is nothing to be modified for the text part, however, figure legends for figures 2 and 3 are not kind for beginners.
A more detailed explanation would be required including meaning of colour difference - red or brown for smads, MAPK, ERK1/2, PI3K/AKT and b-catenin, and red, brown, green, blue for snail, slug, zeb1/2, twist 1/2, and E-cadherin-. Moreover, each upper and lower panel appears to show different concepts for molecular aspect and translational of clinical and molecular aspect, respectively; however, there is not a direct explanation in figure legend (Figure 2).
Answer: We have split Figure 2 in Figure 2A and 2B to facilitate its description within the text. Furthermore, color coding within Figure 2A was unified for signaling molecules, transcription factors, and target genes. The connection between both panels of Figure 2 has been clarified in the revised figure legend.
For figure 3, a direct explanation for each panel would be favourable for better understandings for readers. The colour difference for "personalized therapeutic options" - red, green and blue- might be quite confusing considering each colour has different images in various countries and cultures.
Answer: Multi-modal treatment and definitive RCT have been described more thoroughly in the text and the figure legend to clarify the major differences in therapy, but also to emphasize the repercussions regarding recurrences. The color coding for personalized therapeutic options have been simplified. One single color with a gradient to depict differences across patients have been chosen.
Reviewer 3 Report
The review on EMT transition derived heterogeneity in HNSCC tumors is well written and informative. The reviewed studies are presented clearly and appropriately referenced. Overall, the review is a helpful summary of the available literature and provides insight into possible treatment strategies based on the current data. The only comments for improvement are minor.
- Figure 2 summarizes several key findings and could be split into two figures or labelled A and B. This would aid in referring relevant text to the figures in the manuscript.
- Abbreviations are used throughout the manuscript and it might be helpful to include a table of the abbreviations used. The majority of abbreviations are correctly introduced in the manuscript, but one was not seen for EGFR.
- There are minor grammar and typographical errors in the manuscript. Commas are typically not placed following ‘both’ (line 191, 365, 536) and ‘In the following’ (line 204) is typically followed by section, paragraph, or other place reference. In line 367 ‘cell’ should be ‘cells’ and line 517 ‘will either received’ should be ‘will either have received ‘or ‘will either receive’.
The review on EMT transition derived heterogeneity in HNSCC tumors is well written and informative. The reviewed studies are presented clearly and appropriately referenced. Overall, the review is a helpful summary of the available literature and provides insight into possible treatment strategies based on the current data. The only comments for improvement are minor.
- Figure 2 summarizes several key findings and could be split into two figures or labelled A and B. This would aid in referring relevant text to the figures in the manuscript.
- Abbreviations are used throughout the manuscript and it might be helpful to include a table of the abbreviations used. The majority of abbreviations are correctly introduced in the manuscript, but one was not seen for EGFR.
- There are minor grammar and typographical errors in the manuscript. Commas are typically not placed following ‘both’ (line 191, 365, 536) and ‘In the following’ (line 204) is typically followed by section, paragraph, or other place reference. In line 367 ‘cell’ should be ‘cells’ and line 517 ‘will either received’ should be ‘will either have received ‘or ‘will either receive’.
Author Response
Reviewer 3
The review on EMT transition derived heterogeneity in HNSCC tumors is well written and informative. The reviewed studies are presented clearly and appropriately referenced. Overall, the review is a helpful summary of the available literature and provides insight into possible treatment strategies based on the current data. The only comments for improvement are minor.
- Figure 2 summarizes several key findings and could be split into two figures or labelled A and B. This would aid in referring relevant text to the figures in the manuscript.
- Abbreviations are used throughout the manuscript and it might be helpful to include a table of the abbreviations used. The majority of abbreviations are correctly introduced in the manuscript, but one was not seen for EGFR.
- There are minor grammar and typographical errors in the manuscript. Commas are typically not placed following ‘both’ (line 191, 365, 536) and ‘In the following’ (line 204) is typically followed by section, paragraph, or other place reference. In line 367 ‘cell’ should be ‘cells’ and line 517 ‘will either received’ should be ‘will either have received ‘or ‘will either receive’.
Answer:
- Figure 2 and 3 have both been labelled with A and B to facilitate its description in the text. Furthermore, the figure legends were adapted to be more explanatory.
- A list of abbreviations has been amended and where necessary acronyms have been defined at first use.
- Grammar and typo errors have been addressed.
References
- Tyler, M.; Tirosh, I. Decoupling epithelial-mesenchymal transitions from stromal profiles by integrative expression analysis. Nature communications 2021, 12, 2592, doi:10.1038/s41467-021-22800-1.
- Schinke, H.; Pan, M.; Akyol, M.; Zhou, J.; Shi, E.; Kranz, G.; Libl, D.; Quadt, T.; Simon, F.; Canis, M.; et al. SLUG-related partial epithelial-to-mesenchymal transition is a transcriptomic prognosticator of head and neck cancer survival. Mol Oncol 2021, doi:10.1002/1878-0261.13075.
- Puram, S.V.; Tirosh, I.; Parikh, A.S.; Patel, A.P.; Yizhak, K.; Gillespie, S.; Rodman, C.; Luo, C.L.; Mroz, E.A.; Emerick, K.S.; et al. Single-Cell Transcriptomic Analysis of Primary and Metastatic Tumor Ecosystems in Head and Neck Cancer. Cell 2017, 171, 1611-1624 e1624, doi:10.1016/j.cell.2017.10.044.
Round 2
Reviewer 1 Report
corrections made